# Comparative Insight into Microglia/Macrophages-Associated Pathways in Glioblastoma and Alzheimer’s Disease

**DOI:** 10.3390/ijms25010016

**Published:** 2023-12-19

**Authors:** Jian Shi, Shiwei Huang

**Affiliations:** 1Department of Neurology, Department of Veterans Affairs Medical Center, University of California, San Francisco, CA 94121, USA; 2Department of Neurosurgery, University of Minnesota, Minneapolis, MN 55455, USA

**Keywords:** microglia and macrophage, glioblastoma, TAMs, DAMs, Alzheimer’s disease, NF-κB, miRNAs, TRME2

## Abstract

Microglia and macrophages are pivotal to the brain’s innate immune response and have garnered considerable attention in the context of glioblastoma (GBM) and Alzheimer’s disease (AD) research. This review delineates the complex roles of these cells within the neuropathological landscape, focusing on a range of signaling pathways—namely, NF-κB, microRNAs (miRNAs), and TREM2—that regulate the behavior of tumor-associated macrophages (TAMs) in GBM and disease-associated microglia (DAMs) in AD. These pathways are critical to the processes of neuroinflammation, angiogenesis, and apoptosis, which are hallmarks of GBM and AD. We concentrate on the multifaceted regulation of TAMs by NF-κB signaling in GBM, the influence of TREM2 on DAMs’ responses to amyloid-beta deposition, and the modulation of both TAMs and DAMs by GBM- and AD-related miRNAs. Incorporating recent advancements in molecular biology, immunology, and AI techniques, through a detailed exploration of these molecular mechanisms, we aim to shed light on their distinct and overlapping regulatory functions in GBM and AD. The review culminates with a discussion on how insights into NF-κB, miRNAs, and TREM2 signaling may inform novel therapeutic approaches targeting microglia and macrophages in these neurodegenerative and neoplastic conditions. This comparative analysis underscores the potential for new, targeted treatments, offering a roadmap for future research aimed at mitigating the progression of these complex diseases.

## 1. Introduction

According to recent views, glioblastoma (GBM) and Alzheimer’s disease (AD) are aging-related diseases. GBM, the most common malignant brain tumor, carries a prognosis of a median life expectancy of approximately 15 months post-diagnosis [1]. For the prevalent form of GBM, wild-type isocitrate dehydrogenase (IDH) glioblastoma, the incidence is intrinsically linked to age. However, the mechanisms underlying this association remain a significant and uncharted frontier in GBM research [2]. Based on Alzheimer Association reports, AD is the most common neurodegenerative disease in the elderly, with about 50 million AD patients worldwide, and its incidence rate doubles every five years after the age of 65 years [3,4]. The main clinical phenotypes of AD are cognitive dysfunction, memory loss, and personality changes. Intriguingly, while some research suggests a potential inverse relationship between AD and GBM especially in some gene expression and regulatory pathways [5,6]. Surprisingly, Nelson reported that AD pathology was present in 42% of autopsy cases in patients with GBM, while it was present in 48% for other autopsy cases without GBM, and there were some other similar reports, which, considering the rarity and short survival time of GBM, suggests AD pathology might be underrated in GBM cases [7]. In addition, meta-analyses suggested that AD patients have a higher risk of developing GBM through transcriptomic meta-analyses [8], the underlying mechanisms of which, although not fully understood, were one of the motivations behind our study. Furthermore, it has been shown that the two diseases share some common pathways [2,9]. Therefore, the therapeutic targets for GBM or AD may interact with each other. In addition, we recently found that the bevacizumab (BVZ) treatment for GBM BVZ-non-responsive patients enhanced the age-related gene expression [10]; this result even motivated us to conduct this study.

Microglia and macrophages have recently attracted attention and been shown to play critical roles in diseases, including GBM and AD, after decades of efforts focusing on the tumor cells themselves, neurons, and their cell death. Within the tumor environment of GBM, tumor-associated microglia and macrophages (TAMs) may account for up to 40% of the tumor population, releasing cytokines and growth factors to promote tumor proliferation and survival, angiogenesis, and metastatic progression, and to suppress the function of immune cells [11]. Through their interaction with tumor cells, some TAMs present a unique tumor phenotype of activation, expressing both the M1-and M2-specific markers simultaneously. Their M1 status may regulate the astrocyte malignant transformation in addition to regulating immune responses, while their M2 status may enhance tumor expansion and invasion by increasing angiogenesis and degrading the extracellular matrix (ECM) [12]. Recently, TAMs become an attractive target of GBM studies. It was interesting that the selective removal of resident microglia reduced more tumoral vessels than ablation of the entire TAMs, suggesting that microglia and not peripheral macrophages are the key modulators that promote angiogenesis in GBM [13]. However, recent technological advances in single-cell RNA-sequencing (scRNAseq) methods can be used to reveal key features from highly heterogeneous tumors such as GBM, and using these techniques, microglia-derived TAMs within primary tumors were found to transform into monocyte-derived macrophages in recurrent GBM tumors; therefore, those recurrent TAMs are characterized by more complex immune compartments [14].

Likewise, microglia and macrophages are also key players in the initiation and progression of AD, which is most characterized by senile plaques composed of aggregated amyloid-β (Aβ) peptides and neurofibrillary tangles. Aβ plaques are surrounded by many activated microglia and macrophages [15]. In AD models, these cells continue to secrete pro-inflammation cytokines, including interleukin-1b (IL-1b), p40, inducible nitric oxide synthase (iNOS), and chemokines, but not anti-inflammation cytokines, such as IL-4 and IL-10, suggesting that microglia/macrophages may not shift toward an alternative anti-inflammatory status during pathological processes [16]. Therefore, deficits in phagocytosis and the secretion of inflammatory cytokines are detrimental to neurons and favor Aβ accumulation in AD brains [17]. Consequently, a decrease in macrophage density and shifting the macrophage population toward an anti-inflammatory phenotype was associated with an improved clinical outcome [18].

In this study, we will focus on GBM and AD’s shared pathways involving microglia and macrophages as they are implicated in neuroinflammation, angiogenesis, and immunosuppression in these diseased brains. The pathways discussed here include the nuclear factor kappa-light-chain-enhancer of activated B cells (NF-κB), the regulation of miRNAs, and the triggering receptor expressed on myeloid cells 2 (TREM2) and inflammation; most of the information about them comes from papers published within the past five years. While many pathways associated with these diseases have been extensively reviewed in the literature, specific pathways and their differential regulation could be explored in greater depth. Our focus will be on a more in-depth exploration of the roles of specific cell types in the GBM and AD pathology, particularly examining how these cells influence several of the key pathways mentioned to improve their understanding and bring hope for therapeutic exploration. For the subsequent sections, we will delve into the molecular intricacies of these pathways, exploring their differential regulation and implications in GBM and AD. Specifically, our objectives are to delineate the distinct roles played by microglia and macrophages in GBM and AD and to identify potential molecular targets within these pathways that could lead to innovative therapeutic strategies.

## 2. The Roles of NF-κB in Microglia/Macrophages of GBM and AD

The NF-κB pathway, first identified in 1986 by Sen and Baltimore, plays a pivotal role in the immune response, cell proliferation, and apoptosis. Its molecular architecture includes five DNA-binding members: REL (c-REL), RELA (p65), RELB, NF-κB1 (p50), and NF-κB2 (p52), with the unique attribute of NF-κB2 (p52) lacking transactivation domains. NF-κB signaling encompasses three distinct pathways: canonical, non-canonical, and atypical, each with unique activation mechanisms and cellular responses [19]. Through multiple graphic depictions, the reference clearly presented the traditional pathways and components of NF-κB. The canonical pathway, generally activated by microbial infections or pro-inflammatory cytokines, involves the phosphorylation and subsequent degradation of IκB proteins by the IκB kinase (IKK) complex, releasing p65/p50 NF-κB dimers for nuclear translocation and transcription activation. The non-canonical pathway, selectively activated by receptors like CD40, B-cell-activating factor receptor (BAFF-R), and lymphotoxin beta receptor (LTβR), primarily involves NF-κB2 (p100/p52) proteins and RELB. This pathway initiates with ligand binding, triggering NF-κB-inducible kinase (NIK) to phosphorylate and activate IKK1 (IKKα), leading to p100’s processing into p52 and the subsequent translocation of p52/RELB dimers to the nucleus, thus regulating gene expression differently compared to the canonical pathway. The atypical pathway, which is less well-characterized, can be triggered by DNA-damaging agents independently of IKK, illustrating the versatility and complexity of NF-κB signaling in cellular dynamics [20].

### 2.1. The Roles of NF-κB in TAMs of GBM

NF-κB is commonly upregulated in various GBM cell types, playing a pivotal role in several critical processes. These include not only traditional immune and inflammatory responses, such as the recruiting and aggregation of microglia and macrophages, but also tumor cell survival and anti-apoptosis. These effects are mediated through the upregulation of NF-κB target genes, which encode growth factors, cytokines, and enzymes [19], thereby facilitating tumor cell migration and invasion. Moreover, NF-κB has been implicated in the transition of GBM cells from less aggressive phenotypes, such as the proneural phenotype to the more aggressive mesenchymal phenotype. Beyond these roles, NF-κB’s role in GBM extends to promoting angiogenesis and invasiveness. For instance, the overexpression of Bmi-1 in glioma cells has been linked to increased angiogenesis through the upregulation of vascular endothelial growth factor C (VEGF-C) and NF-κB activation [21]. The experimental use of a non-degradable mutant IκBα, which impedes NF-κB activity, has been observed to curtail the expression of angiogenic molecules and diminish angiogenesis in human glioma xenografts [22]. These findings underscore NF-κB’s multifaceted role in bolstering tumor aggressiveness and progression, rendering it a promising target for therapeutic intervention. All these functions are depicted in Figure 1.

In the context of TAMs within GBM, NF-κB is implicated in resistance to various treatments and the suppression of the immune response. TAM polarization in GBM, encompassing both M1 and M2 states, is significantly influenced by NF-κB. While M1 TAMs promote antitumor responses, M2 TAMs are associated with tumor growth and immune suppression [23]. Predominantly, GBM TAMs display an M2 phenotype with downregulated NF-κB expression, contributing to the immunosuppressive tumor microenvironment (TME) [24]. This includes inhibiting T cell activation and functionality, which is essential for an effective antitumor immune response. Moreover, NF-κB is involved in programmed cell death-ligand1 (PD-L1) expression regulation in GBM cells by initiating PD-L1 gene transcription via promoter binding [25], and indirectly influencing PD-L1 transcriptionally, further contributing to T cell inhibition. The NF-κB signaling modulation of cytokines and chemokines delineates its critical role in reshaping the immune landscape to favor tumor growth.

Strategies aimed at reprogramming TAMs towards an M1 phenotype are thought to be advantageous for patient prognosis. Despite the prevalence of the M2 phenotype during GBM progression, a small fraction of TAMs maintains an M1 status. The presence of necrotic cells within GBM stimulates the expression of TAM chemokines, like the C–C motif chemokine ligand 2/monocyte chemoattractant protein-1 (CL2/MCP-1) and CCL20/MIP-3α (macrophage inflammatory protein 3 alpha), via NF-κB activation, enhancing microglia and macrophage infiltration and attenuating inflammation and apoptosis within the tumor [26], as shown in Figure 1. Innovative treatments like proton irradiation have shown promise in reprogramming M2-like TAMs towards an M1 antitumor phenotype [27], and targeting markers associated with M2-TAMs, such as Arg1 and PD-1, has the potential for reactivating T cell responses and reshaping the TME [28]. Focusing on the NF-κB p50 transcription factor presents another therapeutic avenue. Experiments in which GL261-Luc glioblastoma cells were implanted in both wild-type and p50 knockout (ko) mice resulted in extended survival in the ko mice, characterized by a shift in TAM populations towards pro-inflammatory M1 markers and a threefold increase in tumor-associated CD4 T cells compared to wild-type mice [29]. Ongoing research is critical to elucidate NF-κB’s intricate roles across the spectrum of M1 and M2 macrophage and microglia phenotypes in glioblastoma.

### 2.2. The Roles of NF-κB in Microglia/Macrophges of AD

In AD, the NF-κB pathway is a pivotal mediator in the functional dynamics of microglia and macrophages. The canonical NF-κB pathway, predominantly involving the p65 (RelA) and p50 subunits, is typically associated with the classical M1 activation [30]. This pathway is engaged by pro-inflammatory stimuli, such as lipopolysaccharide (LPS) and cytokines including TNFα and IL-1β, as well as iNOS—a marker commonly attributed to the M1 microglial phenotype. Upon activation, NF-κB orchestrates the microglial release of pro-inflammatory mediators, including IL-1, IL-6, tumor necrosis factor-alpha (TNF-α), interferon-gamma (IFN-γ), and CCL2. These cytokines contribute to neuronal and oligodendrocyte injury, thereby accelerating AD pathogenesis [31]. Consistently, mice lacking the NF-κB1 (p50) subunit (lacking the repressor protein) exhibited enhanced NF-κB activity, leading to more microglia, chronic inflammation, and early onset memory loss, whereas the anti-inflammatory treatment reduced inflammation and restored memory loss [32]. An early event in AD development is microglial activation, where NF-κB signaling in reactive microglia is a key contributor to the sustained inflammation characteristic of the disease. Contrary to the M1 profile, microglia in a traditional ‘M2’ state respond to signals of tissue injury by assuming an amoeboid morphology and migrating to injury sites to phagocytose cellular debris and combat pathogens, thus preserving CNS integrity [33]. Although historically depicted as a binary M1/M2 system—a concept borrowed from macrophage biology—the classification of microglial phenotypes has since evolved. This binary framework is now regarded as overly reductive, particularly within the context of neurodegenerative conditions like AD, and is addressed in depth in a subsequent section.

Nevertheless, therapeutic targeting of the NF-κB pathway in M1 microglia presents a viable strategy for modulating the microglial phenotype balance, offering potential for AD treatment [34,35]. For instance, studies of a microglial BV2 cell culture demonstrated that neuregulin-1 modulates microglial activation and phenotype shifting by inhibiting the NF-κB pathway, likely via an ErbB4-dependent mechanism [35,36]. Similarly, research on Sarcodonin A—a natural compound extracted from mushrooms—highlight its neuroprotective potential by attenuating M1 polarization in microglia, reinforcing its candidacy for AD therapeutics targeting microglial neuroinflammation [35]. Recent findings also underscore the role of Dectin-1, a pattern recognition receptor on microglia, in mediating the inflammatory response to amyloid-beta (Aβ) in AD [37]. The direct interaction between Aβ and Dectin-1 triggers signaling through NF-κB, while Dectin-1 knockout models exhibit a reduction in Aβ-induced microglial activation and inflammation. Moreover, AD brains have shown an upregulation of NF-κB-regulated miRNAs, establishing a gene expression program that correlates with AD pathology, which will be elaborated upon in the following sections. The critical involvement of NF-κB in microglia is further corroborated by the localization of numerous AD risk loci within or adjacent to genes predominantly expressed in microglia, signifying their foundational role in early disease stages. Collectively, these findings bolster the hypothesis that NF-κB targeting in M1 microglia may yield therapeutic advantages in AD.

## 3. miRNA Regulation in Microglia and Macrophages in GBM and AD

MicroRNAs (miRNAs) are pivotal in orchestrating the regulatory mechanisms of microglia and macrophages during the progression of glioblastoma (GBM) and Alzheimer’s disease (AD). These small, non-coding RNA molecules modulate gene expression at the post-transcriptional stage, influencing inflammatory responses and the polarization of microglia/macrophages. Deregulated miRNA expression has been implicated in a spectrum of pathological conditions, including cancer, aging, and neurodegenerative diseases [38,39]. miRNAs can be transported from their origin cells through diverse means—either unbound, protein-associated, or enclosed within extracellular vesicles (EVs) like exosomes—facilitating both local and systemic cell communication [40]. Within the tumor microenvironment (TME) of GBM, miRNAs have a significant role in the dynamic phenotypic shifts of tumor-associated macrophages (TAMs), highlighting their influence on the interplay between TAMs and glioma cells [41]. Similarly, in the context of AD, miRNAs mediate the complex interactions between microglia/macrophages and neuronal cells, with surface molecules, cytokines, exosomal content, and specific signaling pathways playing integral roles in this intercellular communication. Delving into the miRNA-mediated regulation of microglia/macrophages provides a window into understanding GBM and AD pathogenesis and unveils novel avenues for therapeutic intervention in these formidable neurological disorders.

### 3.1. miRNA Regulation in TAMs of GBM

miRNAs are pivotal regulators in glioblastoma (GBM), influencing tumor progression, microenvironmental dynamics, TAM infiltration, and intercellular communication. Specifically, monocyte adhesion to GBM is enhanced by pro-inflammatory factors like TNFα, IFN-γ, and VCAM-1, with the latter’s expression being upregulated by IL-1β in GBM cells, thereby facilitating tumor growth and invasion. Within this framework, the miR-181 family modulates VCAM-1 expression and monocyte adhesion. Notably, reduced miR-181b levels correlate with advanced glioma grades in patients [42]. Regarding TAM infiltration, an inverse relationship exists between miR-340-5p expression and TAM density, as well as M2 polarization in GBM. Low miR-340-5p expression correlates with an increase in ionized calcium-binding adaptor-molecule 1 (Iba1)+ TAMs and CD163+ M2-TAMs. Conversely, higher miR-340-5p expression is associated with fewer M2 TAMs, characterized by dual Iba1+ and CD163+ expression. The downregulation of miR-340-5p in GBM also associates with increased tumor size, recurrence, and diminished survival. Its target, periostin (POSTN)—a gene implicated in TME—recruits TAMs via the integrin avβ3 pathway [43].

Extracellular vesicles (EVs) serve as conduits for the transfer of proteins, nucleic acids, and lipids, particularly miRNAs, which reprogram TAMs and the TME. They also enhance the stem-like characteristics of glioblastoma stem cells (GSCs), and function as intercellular communication vehicles and potential biomarkers. For instance, miR-27b-3p, carried by M2 TAM-derived exosomes, augments GSC tumorigenic traits by fostering growth and upregulating stem cell markers. This is mediated through the direct downregulation of MLL4, a gene integral to GSC functionality. Inhibiting miR-27b-3p in M2-TAMs attenuates the exosomal transfer to GSCs, thereby reducing GSC viability and tumor-promoting activities in vivo. Hence, targeting exosomal miR-27b-3p transfer or the downstream components of the mixed-lineage leukemia 4/positive regulatory domain I element/IL-33 (MLL4/PRDM1/IL-33) signaling axis could impair GSC stem-like properties and enhance the patient prognosis [44]. Furthermore, GSCs secrete exosomal miR-6733-5p, which induces the M2-like polarization of TAMs, bolstering the immunosuppressive milieu of GBM. This polarization, in turn, promotes GSC self-renewal and GBM malignancy, with miR-6733-5p exerting its influence through the insulin-like growth factor 2 mRNA-binding protein 3 (IGF2BP3)-mediated activation of the AKT pathway. Targeting such GSC-released miR-6733-5p offers a potential novel strategy in GBM therapeutics [45], underscoring exosomes’ role in modulating the GBM microenvironment and facilitating communication between GSCs and stromal cells.

The therapeutic potential of miRNA mimics and inhibitors is being explored through GBM treatments, with several candidates currently undergoing preclinical and clinical trials [46]. High-throughput screens have identified miRNAs with cytotoxic effects on GBM cells; notably, miR-1300 induces cytokinesis failure and apoptosis, targeting the oncogene epithelial cell transforming 2 (ECT2) in GBM cells. An inverse correlation between ECT2 mRNA and miR-1300 levels in GBM cells has been documented, suggesting miR-1300’s utility as a GBM treatment option [47]. To date, the therapeutic potential for GBM and other cancers of miR-34a has been investigated several times, as low expression levels of miR-34a in GBM are associated with a high GBM grade and a short survival time [48]. The therapeutic applications of miR-34a for GBM and other cancers have been extensively studied, given that its low expression in GBM correlates with advanced disease and reduced survival. MiR-34a transfection in cancer cells impedes cell cycle progression and diminishes cell viability and invasiveness, acting through targets like Notch 1, myelocytomatosis oncogene (Myc), B cell leukemia/lymphoma 2 (Bcl-2), and CD44 [49]. In the context of temozolomide (TMZ) chemotherapy for GBM, the loss of p53 function can downregulate miR-34a, diminishing Wnt-6 inhibition. MiR-34a, acting as a Wnt pathway inhibitor, can enhance the drug sensitivity of p53-mutant GBM cells and improve survival in xenograft models [50]. Moreover, miR-34a can be modulated by the lncRNA-MUF, which, when upregulated, correlates with a poor GBM prognosis. Knocking down MUF induces miR-34a inhibition and Snail expression upregulation, curtailing the proliferation, migration, and invasion in glioma cells, and increasing the susceptibility to TMZ-induced apoptosis [51]. All related miRNAs are shown in Table 1.

### 3.2. The Role of miRNAs in Regulating Microglia/Macrophages in AD

MicroRNAs (miRNAs) are pivotal in modulating the gene expression and phenotypic polarization of microglia and macrophages in Alzheimer’s disease (AD). Altered miRNA profiles contribute to microglial hyperactivation, sustained neuroinflammation, and aberrant macrophage polarization, correlating with AD pathology [52]. miRNAs regulate proteins critical for amyloid precursor protein processing, tau phosphorylation, synaptic function, and inflammation, thus influencing AD onset and progression [53]. Additionally, they govern microglial clearance of myelin debris, a process vital for resolving inflammation and fostering remyelination. Table 2 encapsulates the miRNAs discussed here.

Therapeutically, miRNAs like miR-22, delivered via exosomes from adipose-derived mesenchymal stem cells, show promise in modulating inflammation and apoptosis in AD models. An Exo-miR-22 treatment in APP/PS1 mice ameliorates the cognitive and motor deficits, dampens microglial activation, and reduces the number of inflammatory cytokines and the neuronal apoptosis rate [54]. In humans, miR-22 is downregulated in AD patients, correlating inversely with inflammation markers. miR-22 mimics in AD models curb pyroptosis, evidenced by the reduction of gasdermin D (GSDMD) and its cleavage products, and notably improve cognitive behavior in mice. In AD animals, giving miR-22 mimic could significantly inhibit pyroptosis, as indicated by the downregulation of GSDMD and p30-GSDMD, after which the release of inflammatory factor was decreased. Most importantly, miR-22 significantly improved the cognitive behavior of AD mice [55].

The deletion of miR-155 in microglia induces an early neurodegenerative microglia (MGnD) phenotype through IFN-γ signaling, which bolsters phagocytosis and leads to cognitive and synaptic improvements in AD mouse models [56]. AD-associated retinal inflammation and vasculopathy are also influenced by miR-155; its deletion mitigates microglial adherence to microvessels and retinal inflammation [57]. Chronic administration of a TNFSF10 antibody in AD models suppresses miR-155 and cytokine expression from microglia, ameliorating AD neuropathology [58]. An inverse relationship exists between miR-17 and NBR1 autophagy cargo receptor (NBR1) in AD patients’ brain microglia; inhibiting elevated miR-17 enhances Aβ clearance and autophagy, underscoring its therapeutic potential. Furthermore, the inhibition of elevated miR-17 in microglia of 5xFAD mice improved Aβ degradation, autophagy, and NBR1 puncta formation in vitro and improved NBR1 expression in vivo [59].

miRNA profiles shift during AD progression, offering potential as biomarkers. AD-induced phagocytic microglia express a specific profile of miRNAs. For AD mice (5xFAD), Aβ-laden microglia were isolated from the brains for RNA sequencing to analyze the transcriptional changes in phagocytic microglia; the specific miRNA profile included the downregulation of miR-7a-5p, miR-29a-3p, and miR-146a-5p, and the upregulation of miR-155-5p and miR-211-5p [60]. In APP/PS1 mice, miR-146a overexpression switched the microglial phenotype, reduced the number of pro-inflammatory cytokines, enhanced the phagocytic function, and reduced the Aβ levels, leading to a reduction in animal cognitive deficits [61], despite conflicting evidence from different models [62]. In extracellular vesicles from AD patients, elevated let-7e levels could serve as diagnostic markers. After treatment with the EVs of AD patients rich with let-7e, IL-6 gene expression was increased in human microglia, suggesting a neuroinflammatory response in microglia to the let-7e [63]. miR-132, which is reduced in AD, is pivotal in maintaining microglial homeostasis and can transition microglia from a disease-associated state to a homeostatic state. For example, in iPSC-derived cultures, miR-132 regulation in the hippocampus leads to a shift from disease-associated microglia (DAM) to homeostatic microglia [64]. From AD patient blood samples, AD showed downregulation of miR-9, miR-21, miR29-b, miR-122, and miR-132 compared to controls. Among these, miR-122 was positively and significantly correlated with several inflammatory factors including granulocyte-macrophage colony-stimulating factor (GM-CSF), INF-a2, IL-1a, IL-8, and major intrinsic protein-1p (MIP-1p) [65]. Moreover, in AD patient blood samples, miR-1908 levels were upregulated, but ApoE levels were reduced. In human macrophage cell line THP-1, giving miR-1908 could inhibit the mRNA and protein levels of ApoE by targeting its 3′untranslated region and inhibiting the ApoE-mediated Aβ clearance [66].

**Table 2 ijms-25-00016-t002:** The effects of miRNAs in microglia/macrophages of AD.

miRNAs	Target	Cell Types	Effects	References
Exo-miR-22	Pyroptotic genes	APP/PS1 mice	Inflammation and apoptosis	[54]
miR-22	GSDMD	Human and AD mice	Inflammation and pyroptosis	[55]
miR-155	IFN-γ, TNFSF10	APP/PS1 mice MGnD microglia	Phagocytotic and microglia deletion	[56,57,58]
miR-17	NBR1	Patients and mice	Aβ degradation	[59]
miR-7a-5p, miR-29a-3p, and miR-146a-5p	Phagocytic mRNAs	Aβ-laden microglia	Microglial phagocytosis	[60]
miR-146a	Change phenotype, TLR	APP/PS1 mice	Inflammation and phagocytosis	[61,62]
Let-7e	IL-6	AD patients	Inflammation	[63]
miR-132	DAM	Human PSCs	Microglia homeostasis	[64]
miR-1908	ApoE	Human blood and macrophage cells	Aβ clearance	[66]

## 4. TREM2 Regulation in Microglia/Macrophages in GBM and AD

The advancing body of research on TREM2 has markedly deepened our comprehension of glioblastoma (GBM) and Alzheimer’s disease (AD), with a particular emphasis on the activation and inflammatory response of microglia/macrophages. TREM2, a member of the immunoglobulin superfamily (Ig-SF), serves as a critical regulator of microglial functions [67]. While the role of TREM2 in GBM has only just begun to be investigated since the TME and TAMs came to the attention of researchers, its roles in other cancer treatments began much earlier [68]. TREM2 has the capacity to bind various anionic ligands—including phospholipids, sulfatides, DNA, and bacterial lipopolysaccharides (LPS)—as well as oligomeric amyloid-beta (Aβ), which is pathognomonic of AD. The binding activities of TREM2 have placed it at the center of immune response regulation in the AD-afflicted brain. Notably, TREM2 variants, such as the R47H mutation, have been implicated in a heightened AD risk, thereby underscoring its integral involvement in the disease’s pathology [69]. TREM2’s interaction with Aβ peptides not only affects microglial activation but also the ensuing neuroinflammatory response, highlighting the intricate nature of AD-associated neuroinflammation. Furthermore, TREM2 transcends simple categorization into phagocytic roles; it intricately modulates DAM phenotypes that present a spectrum of states beyond the traditional binary classification of pro- and anti-inflammatory. Integrating the findings from genetic, molecular, and functional studies illuminates the potential of TREM2-targeted approaches in AD treatment. The ongoing investigation into TREM2 within the AD landscape is paving the way for novel insights into the dynamic between neuroimmune interactions and neurodegeneration, heralding a new era of therapeutic innovation.

### 4.1. The Role of TREM2 in TAMs of GBM

TREM2 is gaining attention as a potential therapeutic target in various cancers, including glioblastoma (GBM). GBM, which is notorious for the presence of excessive apoptotic debris and protein aggregates, shares with TREM2 the substrates meant for phagocytic clearance. Notably, TREM2 expression is elevated in numerous tumors, with its mRNA levels markedly increased in IDH1/2 wild-type GBM when contrasted with normal tissue, indicating a positive correlation between TREM2 expression, the tumor grade, and an unfavorable prognosis in GBM patients [70,71].

A closer examination of the tumor microenvironment (TME) reveals that TREM2 is differentially expressed across various tumor-associated macrophage (TAM) subpopulations. Interestingly, it aligns with the phenotypes often attributed to myeloid-derived suppressor cells (MDSCs) [72]. Single-cell RNA sequencing (scRNA-seq) analyses of the IDH1 wild-type GBM mouse model SB28 demonstrated pronounced TREM2 expression in TAMs (M2-like macrophages and microglia) and nonclassical monocytes—cells known for their anti-inflammatory roles and tissue regulatory functions [73,74]. Conversely, M1-like or IFN-R microglia and certain B cell subsets exhibit a lower TREM2 expression. The highest TREM2 levels are consistently observed in microglial populations within human and murine GBM tumors [75].

Modulating the TREM2 expression has been shown to alter tumor immunotherapy outcomes significantly. In two distinct GBM mouse models, TREM2 knockout resulted in improved overall survival, decelerated tumor progression, and an increase in the number of apoptotic cells within the tumors, hinting at TREM2’s role in TAM-mediated tumor growth inhibition [75]. Since the tumor microenvironment often hinders immunotherapeutic efficacy, targeting TREM2, particularly in conjunction with anti-PD-1 therapy, has been found to suppress tumor growth. A further scRNA-seq post-treatment analysis revealed a reduction in M2-like TAM infiltration and an elevation in M1-like iNOS+ TAMs, which bolster T cell responses [76]. These findings illuminate the potential of TREM2 as a focal point for interventions designed to modulate TAM phenotypes and amplify checkpoint inhibitor effectiveness. The exploration of TREM2 as a therapeutic target in GBM is nascent, yet it is poised to advance rapidly with an expanding comprehension of its fundamental mechanisms that govern the survival, differentiation, and function of microglial and macrophage cells.

### 4.2. The Role of TREM2 Signaling in Microglial Activation in AD

TREM2 is pivotal in regulating the survival, proliferation, and phenotypic transition of microglia/macrophages, particularly in the disease-associated microglia (DAM) phenotype, which plays an integral role in the response to the Alzheimer’s disease (AD) pathology, including amyloid-beta plaque clearance and the modulation of neuroinflammation. The current classification of microglial subtypes in neurodegenerative diseases like AD has evolved beyond the traditional M1/M2 dichotomy to include phenotypes such as IFN-R (Interferon-Responsive) and MHCII (Major Histocompatibility Complex class II), alongside the well-characterized DAM. These subtypes, especially DAM, have functions analogous to the M2 phenotype, like debris clearance and tissue repair support, albeit in a more complex manner. IFN-R microglia are interferon-responsive and may have pro-inflammatory roles, while MHCII microglia are implicated in antigen presentation and inflammatory responses. Cyc-M microglia, however, remain poorly understood. Crucially, the ontogeny and function of these microglial subtypes are TREM2-dependent, and while mostly derived from animal models, the phenotypic markers have been confirmed in human studies using xenografted microglia (xMGs), identified using markers such as purinergic receptor-Y12 (P2RY12), C-X3-C motif chemokine receptor 1 (CX3CR1), and transmembrane protein 119 (TMEM119) for homeostatic subtypes, and APOE, CD11c, and TREM2 for DAM subtypes [77,78].

In the AD brain, TREM2 not only mediates microglial engagement with Aβ plaques but also their phenotypic specialization to DAM. TREM2 interactions with pathological Aβ oligomers activate the microglia, and this is further enhanced through APOE upregulation [79]. DAM cluster around Aβ plaques, while a TREM2 deficiency impairs such clustering and exacerbates the AD pathology, as demonstrated in mouse models [80,81]. TREM2’s role is underscored by the loss of microglial density and activation seen in TREM2-deficient or loss-of-function mutant models, highlighting its necessity for the formation of a protective microglial barrier around plaques [82,83]. An analysis of the microglial transcriptome in mouse models of Aβ plaque accumulation that lacks TREM2 or expresses the R47H TREM2 variant showed that microglia require TREM2 to acquire typical DAM characteristics. At the molecular level, TREM2-deficient or LOF mutant microglia fail to assemble around plaques and to acquire DAM characteristics, in part due to the impaired mTOR pathway activation, resulting in reduced protein synthesis and energy metabolism [84,85]. All the discussed regulatory functions of TREM2 in AD are shown in Figure 2.

Furthermore, TREM2 overexpression in microglia may promote inflammation and early AD pathology progression. Under AD conditions, increased ectodomain shedding of TREM2 by the ADAM metallopeptidase domain 10/17 (ADAM10/17) proteases generates soluble TREM2 (sTREM2), which is detectable in cerebrospinal fluid (CSF). Variations at the H157Y site of TREM2 modulate its cleavage and function, with certain mutations enhancing microglial survival and activity [86]. Nevertheless, TREM2’s influence on AD is multifaceted and varies with disease stage, individual biology, and brain region. In the disease’s early phases, TREM2 deficiency appears to have a protective effect by shifting the cytokine profiles from pro-inflammatory to anti-inflammatory. However, in the later stages, the lack of TREM2 may exacerbate neuroinflammation and disease progression [87], which is consistent with our previous studies in brain injury [88]. These findings suggest a complex, context-dependent impact of TREM2 on AD pathology that necessitates careful consideration in therapeutic strategies.

### 4.3. TREM2 and Microglia Related Therapeutic Implications and Biomarker Potential

TREM2 has emerged as a pivotal target in therapeutic strategies for Alzheimer’s disease (AD) focused on modulating microglial activation and inflammatory responses. As elucidated in earlier discussions, the nuances of TREM2’s role and its genetic variants in microglial function during AD pathogenesis have garnered significant attention. Specifically, TREM2-based therapeutic approaches are gaining traction, inspired by the notion that enhancing TREM2 expression or function could be beneficial, particularly for patients with TREM2 variants or genetic polymorphisms affecting microglial activity. Strategies to augment TREM2 expression may involve thwarting protease-mediated cleavage, while its signaling could be intensified via agonistic antibodies or small molecules that simulate its phospholipid ligands, as shown in Figure 2. Additionally, considering TREM2’s role in sustaining microglial mTOR signaling, metabolic agents designed to improve microglial metabolic health present another promising treatment avenue.

For instance, TREM2 agonist antibodies have demonstrated efficacy in AD animal models. These antibodies stabilize TREM2 on the cell surface, mitigating its shedding, and have been shown to bolster the microglial response, reduce amyloid-beta (Aβ) accumulation, enhance plaque compaction, diminish neurite dystrophy, and ameliorate certain neurological deficits through long-term, high-dose systemic administration [89,90]. However, it is noteworthy that a single administration of these antibodies in advanced-stage AD mice primarily expanded proliferative microglia with a minimal impact on other disease features [91].

The integrity of microglial function is also intimately linked to signaling through the colony-stimulating factor 1 receptor (CSF1R), which is expressed by neurons, astrocytes, and microglia. CSF1R inhibitors, which are capable of depleting microglia in AD models, are currently under investigation as potential AD therapeutics. A chronic administration of a CSF1R inhibitor (PLX5622) in 5xFAD AD mice prevented Aβ plaque formation in areas devoid of microglia and altered the hippocampal gene expression, yet failed to improve neurological deficits [92]. Differing outcomes were observed when microglia were depleted early in the disease’s progression versus in the later stages, implicating microglial presence in plaque maintenance and neurotic dystrophy. Notably, microglial repopulation after inhibitor withdrawal led to plaque compaction, suggesting a neuroprotective role in AD pathology [93].

As for diagnostic utility, soluble TREM2 (sTREM2) levels in cerebrospinal fluid (CSF) are proving valuable for diagnosing AD and tracking cognitive decline. sTREM2 concentrations are initially low in the presymptomatic phase of AD, then peak during the early symptomatic stages—likely marking the onset of microglial activation—and slightly decrease as dementia advances [94]. Correlations between sTREM2 levels and microglial activation linked to Aβ and tau pathologies have been confirmed through positron-emission-tomography (TSPO-PET) imaging in AD patients [95]. Intriguingly, sTREM2 levels have shown divergent predictive values in younger patient cohorts [96] and a positive association with slower cognitive and clinical decline [97]. These findings support the potential of CSF sTREM2 as a biomarker for patient monitoring in AD.

## 5. Discussion

In our manuscript, we explored the shared pathways between glioblastoma (GBM) and Alzheimer’s disease (AD), highlighting NF-κB, miRNA, and TREM2 as key examples, especially focusing on microglia and macrophages in the disease situation. These pathways not only shed light on potential therapeutic targets but also underline the dynamic nature of disease progression influenced by time, genetic predispositions, and disease-specific mechanisms.

Particularly, we delved into the NF-κB signaling pathway, a critical regulator across both GBM and AD. In GBM cases, on the one hand NF-κB upregulation in tumor cells has been linked to the recruitment of microglia and macrophages, alongside promoting angiogenesis and invasiveness. This suggests that NF-κB inhibition could somehow suppress tumor growth and invasion. On the other hand, the tumor microenvironment (TME) is often characterized by TAMs with a predominant M2 phenotype, where NF-κB activity is typically reduced, contributing to immune-suppression. Enhancing NF-κB in the TME could potentially shift TAMs from an M2-like state towards a pro-inflammatory M1-like state, thereby reactivating the immune response against the tumor and potentially improving the efficacy of immune-therapies such as anti-PD1. This dichotomy highlights the need for the strategic modulation of NF-κB signaling within the tumor microenvironment to leverage its dual roles for therapeutic benefit, as delineated in Section 2.1.

In the early stages of AD development, a similar inhibition of NF-κB can attenuate pro-inflammatory microglial activation and mitigate the neurotoxicity of Aβ aggregates, potentially being of benefit to the patients, as discussed in Section 2.2. However, the role of NF-κB in neuronal survival adds a layer of complexity to its modulation. Studies have shown that NF-κB activation plays a part in neuroprotection, with its blockade leading to a loss of this protective effect, particularly in cases where NF-κB is specifically deleted in neurons [98,99]. This highlights a potential conflict in its regulation. In addition, while suppressing NF-κB may inhibit angiogenesis in GBM, a similar inhibitory approach in AD might inadvertently impair neuronal survival. Neurons, after all, require an adequate blood supply, which is regulated in part by angiogenic processes. This dichotomy underscores the need for a nuanced understanding of NF-κB’s role in different neurological contexts and the careful consideration of therapeutic strategies targeting this pathway. Though there is a consistent NF-κB targeting strategy for both diseases, the approaches diverge to accommodate distinct pathophysiological contexts, especially in cellular types and disease contexts.

In these terms, our review has so far elucidated the role of miRNA regulation in GBM and AD, highlighting its potential in therapeutic approaches. Although clinical applications of miRNAs remain nascent, their involvement in critical disease processes—such as neuroinflammation and apoptosis—positions them as promising candidates for therapeutic intervention and diagnostic biomarkers. Given that miRNAs are detectable and stable in various body fluids, including blood, cerebrospinal fluid, and urine, particularly within exosomes, some of them are tissue specific, and they are poised to play a crucial role in mediating information exchange under diverse physiological and pathological states [100]. This unique characteristic position miRNA as a viable candidate for biomarkers and biosensors, potentially revolutionizing the diagnosis and prognosis of disease development and progression. Since 2008, miRNA has been established as a biomarker for diffuse large B-cell lymphoma through patented serum [101], used to diagnose a variety of cancers [102,103], and its use in neurodegenerative diseases, including AD, has begun to be explored [104,105]. For GBM studies, our recent work demonstrates the successful classification and detection of GBM BVZ-responsive subtypes using a combination of three miRNA expressions alongside AI analysis techniques, laying a foundation for personalized medicine approaches in GBM treatment [106]. Furthermore, miRNAs act as key regulators, orchestrating complex gene regulation networks. We discussed, for example, the interplay between these pathways, where miR-34a modulates TREM2 expression via NF-κB signaling—a relationship underscored by a feedback loop where soluble TREM2 (sTREM2) upregulates NF-κB, subsequently affecting miR-34a levels and TREM2 expression [67,76]. This complex interaction could balance microglial activity, influencing both the Aβ pathology and cognitive function, as evidenced through sTREM2’s administration in mice models [107].

The nascent field of miRNA and AI research holds huge untapped potential for GBM and AD therapeutics in the future, yet fully harnessing this potential is challenging due to the intricate nature of miRNA functions. Each miRNA can interact with multiple targets, and these targets often fall under the influence of various miRNAs. This interplay creates a complex regulatory network, making the task of understanding each miRNA’s precise role daunting [108]. The regulation of miRNA expression itself is multifaceted. For instance, long non-coding RNAs (lncRNAs) and circular RNAs (cirRNAs) can act as miRNA sponges, inhibiting their function [109,110], while transcription factors like NF-κB can enhance miRNA expression. This dynamic landscape, including the principles of selective targeting and cooperative interactions, opens a vast field for scientific exploration. Delving into these complexities is key to unlocking the full therapeutic potential of miRNAs in neurodegenerative and neoplastic diseases. In parallel, the integration of advanced AI algorithms has profoundly transformed our approach to interpreting complex imaging data, leading to more precise diagnoses and a deeper understanding of disease progression [111]. Our focus on AI-enhanced PET imaging analysis underscores its potential as a promising tool for identifying novel biomarkers and therapeutic targets [112,113]. This cutting-edge technology marks a significant advancement in visualizing and deciphering complex brain pathologies, propelling GBM and AD research into a new era of innovation.

Contrastingly, TREM2-targeted therapies necessitate opposite directions for GBM and AD, demonstrating the pathway’s intricate role in disease pathology. In GBM, antagonizing TREM2 aims to rejuvenate immune surveillance and to synergize with immunotherapeutic agents like anti-PD-1. Nonetheless, it is important to note that TREM2 is essential for microglia in regulating synaptic engagement during neurodevelopment. Loss of TREM2 function can lead to impaired synaptic maintenance in critical brain regions such as the cortex and hippocampus, potentially resulting in neuronal dysfunction and cognitive impairment [114,115]. These findings suggest that TREM2 may have both tumor-promoting and tumor-suppressive effects in GBM, depending on the specific cellular context and stage of disease progression. In the case of AD, the therapeutic approach leans towards enhancing TREM2 activity through agonistic antibodies, which could bolster DAM functions, aiding Aβ clearance and potentially alleviating neural dysfunction, as seen in Section 4.2. However, the role of TREM2 in AD is not straightforward and can be considered bidirectional. TREM2 loss-of-function variants in humans and genetic defects in animal models have been linked to AD, disrupting microglial aggregation around Aβ plaques and impairing their response to Aβ. This disruption can lead to exacerbated neuronal inflammatory malnutrition. Consequently, the dosage and timing of TREM2 agonistic treatments are critical considerations, underlining the need for precision in therapeutic interventions targeting TREM2 in AD. This nuanced understanding of TREM2’s role across these neurological conditions highlights the challenges in developing targeted therapies and underscores the importance of context-specific strategies.

Our research highlights the potential to influence public health policies, underscoring the importance of early detection and intervention in GBM and AD. Utilizing advanced technological approaches, as explored in our study, could lead to more precise diagnostic and therapeutic strategies, although we know that all new technologies need to be validated over a certain period of research and have their limitations; for example, most scRNA-seq uses smaller sizes compared to traditional bulk RNA-seq. Understanding the disease process is intended to inform healthcare professionals in developing targeted treatments that may reduce the medical burden and provide tangible benefits to patients and their families. These findings connect scientific research to clinical applications and help improve patient care for neurodegenerative and oncological diseases.

## 6. Conclusions

The intersection of microglial and macrophage biology in glioblastoma (GBM) and Alzheimer’s disease (AD) illuminates potential common and contradictory therapeutic avenues, particularly those associated with NF-κB signaling pathways, although these avenues are well-trodden, with an extensive research history. However, the intricacy of these diseases suggests that other pathways, like TREM2, may benefit from a deeper investigative focus and could necessitate divergent strategies that are tailored to disease stage, individual patient factors, and specific disease locales. Beyond direct therapeutic targets, miRNAs emerge as pivotal biomarkers for both GBM and AD, offering a diagnostic and prognostic edge, yet the fundamental principles of their regulatory functions and their complex interaction with multiple targets invite ongoing investigation. Grasping the nuances of shared molecular pathways involving microglia and macrophages is crucial. It holds the promise of pioneering dual-purpose therapeutic strategies in the future. Innovative methodologies, including scRNA-seq, proteomics, and artificial intelligent (AI) analysis, are propelling forward our understanding of these pathways, promising to unravel these shared mechanisms and offering a novel vantage point on treating GBM and AD, particularly by leveraging therapeutic targets associated with tumor-associated macrophages (TAMs) and disease-associated microglia (DAM).

Building upon our findings, future research can explore the potential of targeting specific microglia- and macrophage-associated pathways as a novel therapeutic approach for GBM and AD. This could lead to the development of more personalized medicine strategies, tailoring treatments based on the unique molecular profiles of these diseases. Additionally, further studies might investigate the cross-talk between neuroinflammation and other cellular processes including angiogenesis and aging in GBM and AD, offering insights into new multi-target therapeutic interventions.

## Figures and Tables

**Figure 1 ijms-25-00016-f001:**
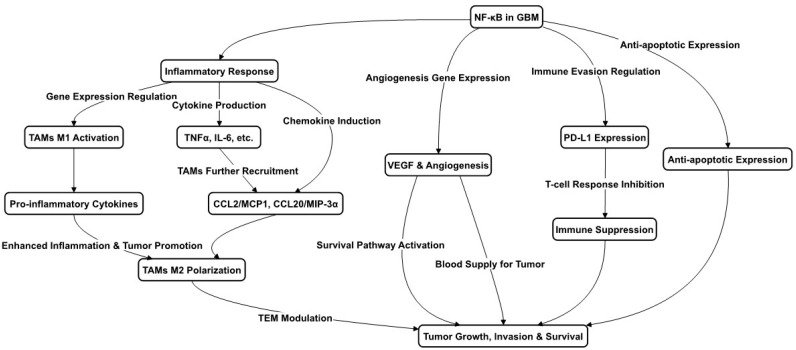
NF-κB Pathways in Glioblastoma. This figure outlines the role of NF-κB in GBM progression, highlighting its involvement in inflammation, immune cell dynamics, and tumor survival. NF-κB drives the transition of macrophages to M1 and M2 types, influencing tumor growth and defense mechanisms. It affects cytokine and chemokine levels, aiding in the recruitment and regulation of cells within the tumor environment. Additionally, NF-κB activity contributes to new blood vessel formation and the suppression of the immune response, promoting tumor resilience.

**Figure 2 ijms-25-00016-f002:**
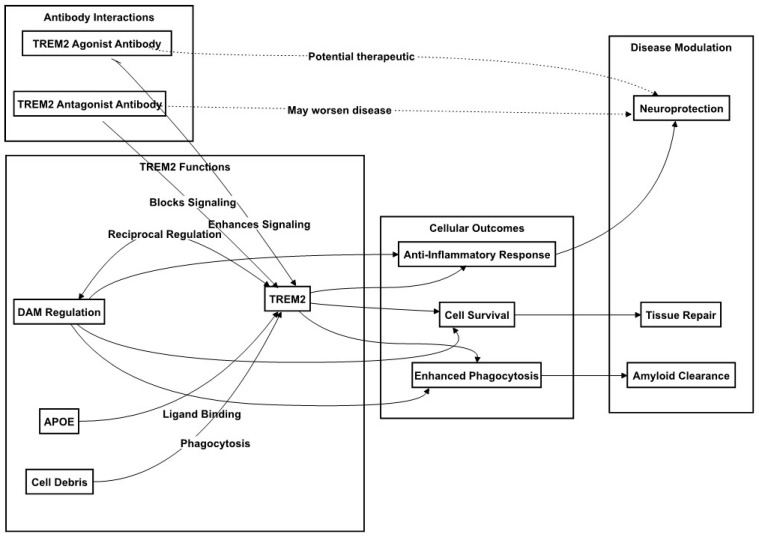
TREM2 Signaling in Alzheimer’s Disease Mechanisms. This figure depicts the critical functions of TREM2 in Alzheimer’s disease, emphasizing its roles in modulating microglial activity, inflammation, and cellular responses. TREM2’s interaction with DAM affects microglial phagocytosis and survival. The figure also shows how TREM2-targeted therapies, via agonistic or antagonistic antibodies, could influence disease outcomes, potentially offering neuroprotection or, conversely, exacerbating disease progression. Solid lines in the figure represent established relationships and pathways, while dotted lines indicate potential therapeutic interventions. The direction of the arrows denotes either the activation or the engagement of the respective components within the signaling pathways.

**Table 1 ijms-25-00016-t001:** The effects of miRNAs in TAMs of GBM and treatment.

miRNAs	Targets	GBM Cell Types	Effects	Reference
miR-181b	VCAM-1	U251 cells	monocyte adhesion	[42]
miR-340-5p	POSTN	Tissues and U251 cells	TAMs recruitment and M2 polarization	[43]
miR-27b-3p	MLL4	GBM tissue	GSCs growth	[44]
miR-6733-5p	IGF2BP3	GSCs and TAMs	TAM polarization and GSCs sternness	[45]
miR-1300	ECT2	GBM cells	Apoptosis	[47]
miR-34a	WNT	xenograft mice	Recover TMZ sensitivity	[50]

## Data Availability

This study didn’t generate new data.

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
