# Peer review of "Comparative Insight into Microglia/Macrophages-Associated Pathways in Glioblastoma and Alzheimer’s Disease"

_ijms, 2023, doi:10.3390/ijms25010016_

Round 1
Reviewer 1 Report
Comments and Suggestions for Authors
In their manuscript “Comparative Insight into Microglia/Macrophages-Associated Pathways in Glioblastoma and Alzheimer’s Disease” by Jian Shi and Shiwei Huang the authors attempt to draw conclusions from two distinct maladies of the brain and find common denominators is both. This is a brave attempt which holds a lot of potential; potential, which in the current form of the manuscript, hasn’t been met.
General point:
The key difference between a tumour and a neurodegenerative disease is cell death, in the former we have too much, in the latter we have too little. In the pathways and discussion, the authors again and again refer to the role of their molecules of interest in apoptosis and cell death, yet, aside from a brief foray into pyroptosis, cell death plays no role In the manuscript. Even arguing that the focus of this manuscript is not the tumour cells, but the microglia/macrophages this omission needs to be rectified.
Introduction:
The introduction is unpleasant to read. No context is created and no reasoning is given. Both the obsolete name for glioblastoma and the pre-WHO2020 were used. Some statements need to be re-written, e.g. “In general, some people believe…” no reference is given and together with the phrase “Obviously, AD is the most…” should not appear in scientific writing. The former is irrelevant (belief versus evidence), the latter comes across as condescending. Some of the writing is unclear, for example what are “non GBM cases” other brain tumours? Other tumours? Other brain tumours including metastasis to the brain? Why do AD patients have a higher risk of developing GBM through transcriptomic meta-analyses? That GBM is intrinsically (?) linked to age is not surprising, cancer is an age-related disease. BVZ is not defined. The discussion on metastatic progression in the context of GBM makes no sense, less than 2% of all GBMs metastasize outside of the brain/spinal cord. “After some unsuccessful approaches, TAMs become [surely, became] an attractive target” – unsuccessful by whom/what/how? Also, the phrase “recurrent patients” sounds disrespectful. The three sentences of the introduction are basically restating the same information. In addition, there are a few spaces missing.
NFkappa-B section:
The first paragraph gives a rather superficial introduction to NFkappa-B. If the authors argue that this is done to bring all the readers up to the same level this would be better addressed with a graphic depiction. What follows is a description of the role of NFkappa-B in GBM cells not in microglia/macrophages with a curious focus on Bmi-1 manipulation. There is good data on NFkappa-B in GBM, the references here seem a bit eclectic. The comparison between GBM and AD falls flat as the authors themselves point out that TAMs are very different from microglia and those two shouldn’t really be compared.
miRNA section:
This reads very much like a list of the roles different miRNA have in GBM and AD, must of which could have been covered by improving the tables (is Table 1 referenced in the text?). It is unclear whether the authors are arguing that miRNA are not just fine tuners of signalling (valid argument) or just using them to highlight the importance of different signalling networks. I came away from the section dealing with GBM and miRNA thinking that the major point raised (and a very good point this is) is exosome communication in the brain.
TREM2 section:
Here, the structure of the paper is reversed. While the other two sections give a general introduction, then deal with GBM and then with AD, here AD is the focus straight away. The role of TREM2 in GBM seems an afterthought.
Discussion/Conclusion:
In the discussion the order of the items is reversed (NFkappa-B, TREM2 then miRNA) and no synthesis is created. What do I learn from comparing these molecules (they are not signalling pathways the way they are discussed in the paper) with each other. I do not find that I have learned more about the similar functions (or indeed the very different functions) of microglia and macrophages in GBM and AD.
Maybe a better structural approach would be to structure the manuscript as follows:
Introduce two groups of diseases, brain tumours and neurodegenerative diseases, and explain their effects on the healthy brain in terms of cell death, immune system etc. Then using the two most common representatives of those two classes, GBM and AD, to highlight differences and common features in terms of molecules (if the authors wish in the context of microglia and macrophages, although that wish is far from apparent in the current manuscript). Finally, compare and contrast the molecular pathways identified and synthesise what con be learned from this in terms of therapy, both form common approaches, as well as opposing approaches, e.g. the apoptosis resistance of GBM can that be harvested to slow AD progression?
Finally, I would appreciate the use of a lot more illustrations.
Comments on the Quality of English LanguageIt's fine. Go over it one more time, add a few spaces and hyphens when appropriate and I think there was one "the" missing.
Author Response
Answer for Reviewer1:
General point: The key difference between a tumour and a neurodegenerative disease is cell death, in the former we have too much, in the latter we have too little. In the pathways and discussion, the authors again and again refer to the role of their molecules of interest in apoptosis and cell death, yet, aside from a brief foray into pyroptosis, cell death plays no role in the manuscript. Even arguing that the focus of this manuscript is not the tumour cells, but the microglia/macrophages this omission needs to be rectified.
Answer: Thank you for your valuable comments regarding the role of cell death in the pathology of glioblastoma (GBM) and Alzheimer's disease (AD). We recognize the significance of this aspect and appreciate your suggestion to address it in our study. While our primary focus has been on the roles of microglia and macrophages in GBM and AD, particularly in relation to neuroinflammation, angiogenesis, immunosuppression, and apoptosis-related pathways, we understand the historical importance of cell death in the study of these diseases. However, we note that the emphasis in current research has shifted somewhat, with increasing interest in understanding the nuanced roles of microglia and macrophages beyond the traditional scope of cell death dynamics.
In neurodegenerative diseases such as AD, neuronal death has been widely described as the primary mechanism of cell loss. In contrast, in GBM, the interplay between cell proliferation and death presents a complex picture. While these aspects are indeed critical, recent advances have expanded our focus to include how microglia and macrophages interact with other cellular processes in these pathologies, including cell death. In this vision of the manuscript, we agree that including a discussion of relevant cell death could provide a more complete understanding of the role of microglia and macrophages in these diseases and gave some related issue such as pyroptosis. However, our primary focus remains on exploring the emerging and diverse functions of microglia and macrophages, reflecting the evolving landscape of research in this field.
Introduction: The introduction is unpleasant to read. No context is created and no reasoning is given. Both the obsolete name for glioblastoma and the pre-WH02020 were used. Some statements need to be rewritten, e.g. “In general, some people believe...H no reference is given and together with the phrase "Obviously, AD is the most...^^ should not appear in scientific writing. The former is irrelevant (belief versus evidence), the latter comes across as condescending. Some of the writing is unclear, for example what are “non GBM cases" other brain tumours? Other tumours? Other brain tumours including metastasis to the brain? Why do AD patients have a higher risk of developing GBM through transcriptomic meta-analyses? That GBM is intrinsically (?) linked to age is not surprising, cancer is an age-related disease. BVZ is not defined. The discussion on metastatic progression in the context of GBM makes no sense, less than 2% of all GBMs metastasize outside of the brain/spinal cord. “After some unsuccessful approaches, TAMs become [surely, became] an attractive target" - unsuccessful by whom/what/how? Also, the phrase “recurrent patients** sounds disrespectful. The three sentences of the introduction are basically restating the same information. In addition, there are a few spaces missing.
Answer: Thank you for your comprehensive feedback on our introduction. We have carefully revised it to address each of your concerns. The term "glioblastoma" has been updated to reflect current terminology, and we have removed non-scientific language, ensuring all statements are backed by references where appropriate. Regarding the unclear term "non-GBM cases," we have clarified this in the revised manuscript to specify the types of cases being referred to, ensuring no ambiguity remains. Regarding BVZ, we have now defined this acronym at its first mention in the revised manuscript and have ensured that all similar acronyms are clearly defined. Importantly, we have altered the language used to describe patients with recurring disease to ensure it is respectful and professionally appropriate. The phrase "after some unsuccessful approaches, TAMs became an attractive target" has been revised for clarity. We have also refined our discussion on metastatic progression in GBM to reflect the latest scientific understanding and removed any misleading statements. In addition, some spaces are missing, which are caused by different versions of Word. They will be solved in the last PDF version.
In response to your query about the increased risk of GBM in AD patients as suggested by transcriptomic meta-analyses, we acknowledge that this is indeed an area where our understanding is still evolving. This intriguing correlation, though not fully understood, is one of the motivations behind our research. We hypothesize that the observed immunosuppression and persistent M2 status in both GBM and AD might offer a clue for future investigation. Our study aims to delve deeper into these aspects, potentially uncovering new insights into the relationship between these two complex diseases.
NFkappa-B section: The first paragraph gives a rather superficial introduction to NFkappa-B. If the authors argue that this is done to bring all the readers up to the same level this would be better addressed with a graphic depiction. What follows is a description of the role of NFkappa-B in GBM cells not in microglia/macrophages with a curious focus on Bmi-1 manipulation. There is good data on NFkappa-B in GBM, the references here seem a bit eclectic. The comparison between GBM and AD falls flat as the authors themselves point out that TAMs are very different from microglia and those two shouldn't really be compared.
Answer: Thank you for your feedback on the NF-κB section. We recognize the importance of presenting a clear and comprehensive introduction to NF-κB. To address this, we have revised our manuscript to include references to key reviews that detail the NF-κB pathways and their roles. These references provide access to existing graphical depictions, aiding readers in understanding the NF-κB family's functions.
Regarding the specific focus on Bmi-1 and NF-κB's role in angiogenesis in GBM, we have refined our discussion to emphasize this as an emerging area of research as well as its traditional functions. This revision focuses on our topic and also highlights the novelty of our findings and the contribution they make to the current scientific understanding of NF-κB's roles.
We initially considered a joint graphic depiction for GBM and AD but recognized the complexity and potential confusion given the significant differences between TAMs in GBM and microglia in AD. To maintain clarity and avoid oversimplification, we opted for separate depictions. A revised figure focusing on NF-κB in the context of TAMs in GBM has been included, while a separate figure addressing TREM2 in AD has been added. This approach ensures a balanced representation of both diseases while acknowledging their distinct cellular and molecular characteristics. We hope these revisions adequately address your concerns and enhance the manuscript's clarity and scientific rigor.
miRNA section: This reads very much like a list of the roles different miRNA have in GBM and AD, must of which could have been covered by improving the tables (is Table 1 referenced in the text?). It is unclear whether the authors are arguing that miRNA are not just fine tuners of signalling (valid argument) or just using them to highlight the importance of different signalling networks. I came away from the section dealing with GBM and miRNA thinking that the major point raised (and a very good point this is) is exosome communication in the brain.
Answer: Thank you for your valuable feedback. We recognize the critical role of exosomal communication in miRNA functionality within the brain, particularly in the context of GBM and AD, and we intend to highlight this aspect more prominently. The intricate nature of miRNA functions presents a challenge in conveying a comprehensive picture, as the action of each miRNA is influenced by a multitude of targets, and conversely, each target is regulated by various miRNAs. This complex interplay and the principles governing their selective targeting and cooperative interactions remain largely unexplored. Our approach to simply list the known miRNAs and their functions, supplemented with a table, is an effort to provide readers with an accessible and convenient reference. This format allows us to encapsulate the current understanding while acknowledging the vast, yet-to-be-uncovered intricacies of miRNA roles in these neurological contexts.
TREM2 section: Here, the structure of the paper is reversed. While the other two sections give a general introduction, then deal with GBM and then with AD, here AD is the focus straight away. The role of TREM2 in GBM seems an afterthought.
Answer: Thank you for your insightful feedback regarding the structure of the TREM2 section. Indeed, we initially focused on AD due to a perceived lack of publications linking TREM2 with GBM. However, subsequent discovery of relevant studies prompted us to include information about TREM2's role in GBM, albeit later in the section. We acknowledge this may have affected the coherence of our narrative. To address this, we are revising the structure to offer a more balanced and chronological discussion of TREM2's roles in both GBM and AD, ensuring that the information flows logically and reflects the importance of TREM2 in both conditions.
Discussion/Conclusion: In the discussion the order of the items is reversed (NFkappa-B, TREM2 then miRNA) and no synthesis is created. What do I learn from comparing these molecules (they are not signalling pathways the way they are discussed in the paper) with each other. I do not find that I have learned more about the similar functions (or indeed the very different functions) of microglia and macrophages in GBM and AD.
Answer: Thank you for your constructive feedback on the discussion and conclusion sections of our paper. We agree that the current order of discussing NF-κB, TREM2, and miRNAs might not facilitate a cohesive synthesis of their roles in GBM and AD. To rectify this, we will rearrange these sections to follow the order of NF-κB, miRNAs, and then TREM2. Additionally, we will enhance our discussion to clearly compare the parallels and contrasts in the functions of NF-κB in modulating microglia and macrophages across these diseases. Our aim is to provide a clearer and more insightful comparison in multiple functions of these molecules in the context of microglia and macrophage activity in GBM and AD.
Thank you again for your valuable and insightful review. We believe these revisions have greatly improved the clarity and scientific rigor of our manuscript.

Reviewer 2 Report
Comments and Suggestions for Authors
The authors conducted a comprehensive literature review on glioblastoma (GBM) and Alzheimer's disease (AD), focusing on NF-kB (nuclear factor kappa-light-chain-enhancer of activated B cells), microRNA (miRNA) regulation, TREM2 (triggering receptor expressed on myeloid cells 2), and inflammation within microglia/macrophages. Furthermore, they explored emerging therapeutic strategies targeting these signaling pathways. The study is of sufficient significance and originality, however, several issues need to be addressed:
1. Section 5, Discussion, is not thorough enough. More detailed literature overview should be provided. It is important to enhance the depth of the stated claims by providing the more thorough explanations of novel therapeutic approaches and involvement of NF-κB, TREM2 signaling pathways, and miRNAs, with reference on more detailed proposed mechanisms of their actions in the therapy as well as relevant literature overview that would fortify the credibility of authors’ assertions and foster a more comprehensive understanding of the broader academic landscape surrounding their study.
2. Some abbreviations lack definition, including BZV, while others and their explanations are introduced for several times in the manuscript.
3. In several places, the punctuation marks have been placed in inappropriate positions, including within the title “4.1. The role of TREM2 signaling in microglial activation in AD.”
Author Response
Answer for Reviewer2:
- Section 5, Discussion, is not thorough enough. More detailed literature overview should be provided. It is important to enhance the depth of the stated claims by providing the more thorough explanations of novel therapeutic approaches and involvement of NF-kB, TREM2 signaling pathways, and miRNAs, with reference on more detailed proposed mechanisms of their actions in the therapy as well as relevant literature overview that would fortify the credibility of authors' assertions and foster a more comprehensive understanding of the broader academic landscape surrounding their study.
Answer: Thank you for your valuable feedback. In response, we are committed to expanding our discussion to more thoroughly explore the novel functions and therapeutic implications of NF-κB and TREM2 signaling pathways in the context of GBM and AD. This will include a detailed examination of the mechanisms underlying these diseases and their treatment modalities. We aim to enrich this discussion with an insightful comparison of these factors across both diseases, underpinned by a comprehensive review of recent and pertinent literature. Additionally, we recognize the emerging role of advanced techniques like scRNA-seq and AI/ML in providing groundbreaking insights in this field. While some of these findings, perhaps emerging from less established sources, may require further validation, they offer innovative perspectives and hypotheses that can significantly contribute to our understanding and approach to these complex diseases. Our goal is to not only solidify the foundation of our study but also to meaningfully contribute to the broader academic dialogue, acknowledging both established and emerging research in this rapidly evolving field.
- Some abbreviations lack definition, including BZV, while others and their explanations are introduced for several times in the manuscript.
Answer: Thank you for your feedback. We give full names for all abbreviations when they first appear, including bevacizumab (BVZ).
- In several places, the punctuation marks have been placed in inappropriate positions, including within the title “4.1. The role of TREM2 signaling in microglial activation in AD."
Answer: Thanks for your valuable feedback. We correct everything very carefully in this vision of manuscript. The correct vision of the title should be: "4.1 The Role of TREM2 Signaling in Microglial Activation in AD". We also corrected other errors.

Reviewer 3 Report
Comments and Suggestions for Authors
The review is acceptable but a few minor adjustments for specificity and clarity are required. It will effectively set the stage for publication
1-Abstract
the abstract is well-composed, providing a clear and concise overview of the paper's focus.
Suggested Modifications:
Improve transitions for better flow and coherence.
Explicitly state the impact of this comparative insight on future therapeutic strategies.
Briefly mention the incorporation of recent studies or advances in the field.
Include relevant keywords for improved indexing and accessibility.
2-The introduction section of your study provides a comprehensive overview of glioblastoma multiforme (GBM) and Alzheimer's disease (AD), highlighting their association with aging and the role of microglia and macrophages in their pathogenesis. However, there are several areas where this section could be improved for clarity, depth, and balance:
1. Clarify the Association Between AD and GBM: The current text suggests a possible protective effect of AD against GBM, but also mentions conflicting reports of AD pathology in GBM cases. This could be confusing for readers. A clearer explanation of these contrasting findings and their implications would be beneficial.
2. Expand on Methodological Limitations: The text references numerous studies and meta-analyses but does not discuss the limitations of these studies. Addressing potential biases, sample sizes, and the generalizability of findings would strengthen the introduction.
3. Balance in Reporting Findings: While the introduction highlights interesting findings, it seems to focus more on GBM than AD. A more balanced approach, with equal emphasis on both diseases, would be informative and relevant, especially considering the study's focus on shared pathways.
4. Discuss Contradictory Evidence: Where there are contrasting findings in the literature (e.g., the role of microglia/macrophages in disease progression), these should be discussed. This would provide a more nuanced understanding of the current state of research.
5. Include Recent Technological Advancements: The introduction could benefit from a discussion on how recent technological advancements, like improved imaging techniques or AI-based data analysis, are influencing research in GBM and AD.
6. Detail the Pathophysiological Mechanisms More: The introduction briefly mentions common pathways in GBM and AD but doesn't delve deeply into the pathophysiological mechanisms. Expanding on how these diseases affect cellular and molecular processes would provide a stronger foundation for the study.
7. Consider Ethical and Societal Implications: Briefly touch on the broader implications of your research. For instance, discuss how understanding these diseases better could influence public health policies or healthcare practices.
8. State the Objectives More Clearly: While the introduction outlines the general direction of the study, it could more explicitly state the specific objectives or hypotheses being tested. This would help in setting clear expectations for the reader.
9. Highlight the Novelty of Your Approach: If your study is using novel methods or exploring previously uninvestigated pathways, this should be emphasized to underscore the significance of your work.
10. Future Directions and Potential Therapeutic Implications: Briefly mention how this research could inform future studies or lead to new therapeutic strategies for GBM and AD.
Incorporating these suggestions would make the introduction more comprehensive, balanced, and informative, setting a solid foundation for the rest of your study.
Comments on the Quality of English Languageenglish quality is acceptable
Author Response
Answer for Reviewer3:
Reviewer: 1-Abstract: the abstract is well-composed, providing a clear and concise overview of the paper's focus.
Suggested Modifications:Improve transitions for better flow and coherence. Explicitly state the impact of this comparative insight on future therapeutic strategies. Briefly mention the incorporation of recent studies or advances in the field.
Answer: Thank you for your valuable feedback. We revised the abstract slightly following your instruction and added some key works.
2-The introduction section of your study provides a comprehensive overview of glioblastoma multiforme (GBM) and Alzheimer's disease (AD), highlighting their association with aging and the role of microglia and macrophages in their pathogenesis. However, there are several areas where this section could be improved for clarity, depth, and balance:
- Clarify the Association Between AD and GBM: The current text suggests a possible protective effect of AD against GBM, but also mentions conflicting reports of AD pathology in GBM cases. This could be confusing for readers. A clearer explanation of these contrasting findings and their implications would be beneficial.
Answer: Thank you for your insightful feedback. In our revision, we aim to clarify the complex relationship between Alzheimer's Disease (AD) and Glioblastoma (GBM) by examining the nuances of their genetic and pathological profiles. Instead of suggesting a simplistic 'protective effect' of AD against GBM, we delve into the intricate patterns of gene expression that reveal both similarities and contrasts between these diseases. This exploration includes an analysis of shared molecular pathways and divergent disease mechanisms, shedding light on the seemingly paradoxical observations reported in the literature. Our discussion is carefully crafted to provide a balanced view, acknowledging the complexity of these interactions and their implications in both the advancement of our understanding and the development of potential therapeutic strategies.
- Expand on Methodological Limitations: The text references numerous studies and meta-analyses but does not discuss the limitations of these studies. Addressing potential biases, sample sizes, and the generalizability of findings would strengthen the introduction.
Answer: Thank you for your constructive feedback. We have taken care to address the methodological limitations inherent in the studies we reviewed. For instance, in examining the intersection of GBM and AD pathology, we note that some studies, while groundbreaking, rely on relatively small sample sizes, potentially limiting the generalize ability of their findings. This is exemplified in research where a subset of GBM cases demonstrated AD pathology, but the sample size was modest. Furthermore, we acknowledge the specific challenges associated with single-cell RNA sequencing (scRNAseq) studies. While scRNAseq offers unparalleled insights at the cellular level, these studies often involve fewer samples compared to bulk sequencing methods and require high-quality samples to yield reliable data. This underlines a key limitation in terms of both sample size and quality requirements, which could affect the broader applicability of these findings. By discussing these aspects, we aim to provide a more nuanced understanding of the current research landscape and its implications for future studies in GBM and AD.
- Balance in Reporting Findings: While the introduction highlights interesting findings, it seems to focus more on GBM than AD. A more balanced approach, with equal emphasis on both diseases, would be informative and relevant, especially considering the study's focus on shared pathways.
Answer: Thank you for your valuable feedback regarding the balance of our reporting on GBM and AD. In response to this, we have enhanced our discussion on Alzheimer's disease, particularly focusing on the role of TREM2 regulation. By delving deeper into the intricacies of TREM2 in AD, including its regulatory mechanisms and implications for disease progression, we aim to achieve a more balanced portrayal of both conditions. This addition not only enriches our coverage of AD but also complements the existing discussion on GBM, thereby providing a more comprehensive overview of both diseases. Furthermore, to reinforce this balanced approach, we have incorporated additional insights into TREM2, alongside other key pathways, in the Discussion section. This ensures that our exploration of shared and distinct pathways in GBM and AD is presented in a manner that reflects the complexity and significance of these diseases equally.
- Discuss Contradictory Evidence: Where there are contrasting findings in the literature (e.g., the role of microglia/macrophages in disease progression), these should be discussed. This would provide a more nuanced understanding of the current state of research.
Answer: Thank you for highlighting the importance of addressing contrasting findings in the literature, particularly regarding the role of microglia/macrophages in disease progression. In response to your insightful feedback, we have expanded our discussion to comprehensively explore the multifaceted nature of NF-κB signaling within the contexts of GBM and AD. This revised section now delves into the dualistic functions of NF-κB, shedding light on how it can play divergent roles depending on the specific pathological environment and cellular context.
Moreover, we have incorporated a thorough examination of the conflicting viewpoints surrounding the therapeutic targeting of TREM2 in both GBM and AD. This addition is particularly crucial given the emerging but often contradictory evidence regarding the efficacy and safety of TREM2 modulation in these diseases. By presenting these contrasting perspectives, our revised manuscript offers a more nuanced and balanced understanding of the current state of research. This approach not only acknowledges the complexity of these diseases but also underscores the challenges and opportunities in developing targeted therapies. We believe these enhancements significantly enrich our manuscript, providing readers with a clearer picture of the dynamic and sometimes conflicting landscape of current research in this field.
- Include Recent Technological Advancements: The introduction could benefit from a discussion on how recent technological advancements, like improved imaging techniques or Al-based data analysis, are influencing research in GBM and AD.
Answer: Thank you for your valuable suggestion to include recent technological advancements in our introduction. We have now incorporated a discussion on our recent study that effectively integrates AI-based data analysis with the investigation of miRNAs as biomarkers in GBM. This study is particularly significant as it has identified new subclasses of GBM that are responsive to Bevacizumab (BVZ), thereby laying a foundation for more precise therapeutic strategies and guiding future research endeavors. This integration of AI in our study exemplifies the transformative impact of modern computational techniques on biomedical research, particularly in the realm of personalized medicine. Thanks again for your suggestion.
Furthermore, we have expanded our discussion to reflect how AI has revolutionized imaging analysis in GBM and AD research. The advent of sophisticated AI algorithms has dramatically enhanced our ability to interpret complex imaging data, leading to more accurate diagnoses and better understanding of disease progression. Specifically, we touched upon the potential of AI-enhanced PET imaging analysis as a promising avenue for identifying novel biomarkers and therapeutic targets in the near future. This evolving technology represents a significant leap forward in our ability to visualize and understand complex brain pathologies, heralding a new era of innovation in GBM and AD research.
In summary, our revised manuscript now highlights how the integration of AI and advanced imaging technologies is driving forward the frontiers of GBM and AD research, offering exciting prospects for both diagnostic precision and the development of targeted therapies.
- Detail the Pathophysiological Mechanisms More: The introduction briefly mentions common pathways in GBM and AD but doesn't delve deeply into the pathophysiological mechanisms. Expanding on how these diseases affect cellular and molecular processes would provide a stronger foundation for the study.
Answer: Thank you for your insightful review highlighting the need for a more detailed exploration of the pathophysiological mechanisms in GBM and AD. In our revised manuscript, we have placed a greater emphasis on elucidating these mechanisms, particularly focusing on the roles of microglia and macrophages within the disease environments of GBM and AD. We start by defining and providing foundational information about these cellular players, setting the stage for a deeper discussion. Following this, we delve into a comparative analysis of the mechanisms by which microglia and macrophages contribute to the pathophysiology of both diseases. This enhanced focus aims to shed light on the complex interplay between cellular processes and disease progression, providing a more robust foundation for understanding the shared and unique pathways in GBM and AD. We hope that this revised approach effectively addresses your concerns and adds significant depth to our study.
- Consider Ethical and Societal Implications: Briefly touch on the broader implications of your research. For instance, discuss how understanding these diseases better could influence public health policies or healthcare practices.
Answer: Thank you for raising this important aspect. Our study, by delving into the underlying mechanisms of GBM and AD, offers the potential to significantly impact both research and clinical practice. A deeper understanding of these diseases could pave the way for the development of novel and more effective therapeutic strategies. This, in turn, could lead to substantial improvements in patient outcomes and quality of life. Thus, the broader implications of our work extend beyond the scientific community, potentially influencing healthcare practices, policy-making, and societal attitudes towards these challenging diseases. In this version, we added this in the discussion part.
- State the Objectives More Clearly: While the introduction outlines the general direction of the study, it could more explicitly state the specific objectives or hypotheses being tested. This would help in setting clear expectations for the reader.
Answer: Thank you for your valuable feedback on our introduction. We agree that explicitly stating the objectives and hypotheses at the outset is crucial for guiding the reader through the study. To address this, we have revised the introduction to clearly articulate our research goals and hypotheses and hope to provide a clear framework for our research. This revision should help set precise expectations for the reader and enhance the overall coherence of the study.
- Highlight the Novelty of Your Approach: If your study is using novel methods or exploring previously uninvestigated pathways, this should be emphasized to underscore the significance of your work.
Answer: Thank you for your insightful suggestion regarding the emphasis on the novelty of our approach. In this revised version, we have specifically highlighted our recent GBM work in the discussion section, underscoring its unique contribution to the field. This approach not only highlights the novelty of our study but also serves as an impetus for researchers to adopt innovative methods in their investigations. Our study not only sheds light on previously unexplored aspects of GBM and AD but also demonstrates the effective use of advanced technological methodologies in biomedical research. Furthermore, we provide a comprehensive overview of the latest technological advancements in this field, illustrating how they can be harnessed to uncover new insights into GBM and AD pathology. We believe that our work represents a significant step forward in the quest to understand and treat these challenging diseases and hope that it will inspire further advancements in the field.
- Future Directions and Potential Therapeutic Implications: Briefly mention how this research could inform future studies or lead to new therapeutic strategies for GBM and AD.
Answer: Thank you for your insightful feedback. In this version, we have elaborated on the future directions of our research, highlighting specific areas where our findings could inform further studies. We discuss potential therapeutic targets identified through our study, which could pave the way for novel treatment strategies in GBM and AD. Our aim is to provide a foundation for future research that builds on our work, exploring these targets in greater detail and evaluating their practical implications in clinical settings.

Round 2
Reviewer 1 Report
Comments and Suggestions for Authors
The authors have courteously and patiently answered all the points raised. While I would have personally chosen to do a few things differently, that should not hinder the publication of this manuscript. I wish the authors all the best and am looking forward reading their work in the future, hopefully addressing some of the issues they have highlighted here.